# Photothermally-Heated Superparamagnetic Polymeric Nanocomposite Implants for Interstitial Thermotherapy

**DOI:** 10.3390/nano12060955

**Published:** 2022-03-14

**Authors:** Ivan B. Yeboah, Selassie W. K. Hatekah, Abu Yaya, Kwabena Kan-Dapaah

**Affiliations:** 1Department of Biomedical Engineering, School of Engineering Sciences, University of Ghana, Legon, Accra P.O. Box LG 77, Ghana; iv_yeb@yahoo.com (I.B.Y.); hwk.selassie@gmail.com (S.W.K.H.); 2Department of Materials Science and Engineering, School of Engineering Sciences, University of Ghana, Legon, Accra P.O. Box LG 77, Ghana; ayaya@ug.edu.gh

**Keywords:** nanocomposite implants, interstitial heating, photothermal therapy, near-infrared, cancer

## Abstract

Photothermally-heated polymer-based superparamagnetic nanocomposite (SNC) implants have the potential to overcome limitations of the conventional inductively-heated ferromagnetic metallic alloy implants for interstitial thermotherapy (IT). This paper presents an assessment of a model SNC—poly-dimethylsiloxane (PDMS) and Fe${}_{3}$O${}_{4}$ nanoparticles (MNP)—implant for IT. First, we performed structural and optical characterization of the commercially purchased MNPs, which were added to the PDMS to prepare the SNCs (MNP weight fraction =10 wt.%) that were used to fabricate cubic implants. We studied the structural properties of SNC and characterized the photothermal heating capabilities of the implants in three different media: aqueous solution, cell (in-vitro) suspensions and agarose gel. Our results showed that the spherical MNPs, whose optical absorbance increased with concentration, were uniformly distributed within the SNC with no new bond formed with the PDMS matrix and the SNC implants generated photothermal heat that increased the temperature of deionized water to different levels at different rates, decreased the viability of MDA-MB-231 cells and regulated the lesion size in agarose gel as a function of laser power only, laser power or exposure time and the number of implants, respectively. We discussed the opportunities it offers for the development of a smart and efficient strategy that can enhance the efficacy of conventional interstitial thermotherapy. Collectively, this proof-of-concept study shows the feasibility of a photothermally-heated polymer-based SNC implant technique.

## 1. Introduction

The ferromagnetic implant (thermoseed) technique is a minimally invasive modality for interstitial thermotherapy (IT) involving two main steps. First, an array of the implants (small rods with a diameter of ≈1 mm and length between a millimeter to a few centimeters [1,2]) are placed in the target tissue in the subject using either open surgery or image-guided percutaneous insertion. Then, the subject is placed within an extracorporeal alternating magnetic field that causes the implants to generate heat that is transferred to the tumor by conduction. The feasibility and effectiveness of the technique have been explored in several studies ranging from fundamental research through to clinical studies [1,2,3,4,5,6,7]. The technique has been described as having features that are desirable for interstitial therapy including the thermal self-regulating capability of the thermoseeds coupled with the elimination of the need for physical contact between thermoseeds and the excitation field [1,2,3]. However, issues related to biocompatibility and corrosiveness of the metallic alloy implants as well as the need to use a high number of implants to achieve therapeutic temperatures have affected effectiveness and precluded their full clinical use [8,9]. Therefore, there is a need for novel strategies that can overcome the issues associated with the thermoseed technique.

Minimally invasive photothermal therapy has recently received a lot of attention because of its ability to penetrate deep tissue while also reducing the impact of non-selective cell death when combined with nanotechnology, so-called nanotechnology-assisted photothermal therapy (N-PTT) [10,11,12,13]. Recent advances in laser delivery equipment technology allow for the development of more reliable, secure, and cost-effective interstitial strategies [10,11,14]. N-PTT relies on the ability of nanoparticles (NPs) to absorb and convert near-infrared (NIR) radiation to heat for therapeutic purposes. Preferred NP materials used for N-PTT include metallic (e.g., Au, Ag, Cu), graphene and carbon-nanotubes due to the opportunity to manipulate their localized surface plasmon resonance (LSPR) to enhance their absorption capabilities [14], however, issues related to NP biocompatibility and stability have precluded their clinical use [15]. Unlike their metallic counterparts, iron-oxide (Fe${}_{3}$O${}_{4}$ and γ−Fe${}_{2}$O${}_{3}$) NPs have been approved by the FDA for clinical magnetic thermotherapy [16,17]. Their intrinsic ability of generate heat within a short time under NIR irradiation as well as drive high barrier reactions makes them attractive for N-PTT [13,18,19,20]. Recently, these ceramic NPs have been demonstrated in single (photothermal only) or multimodal (simultaneous photothermal & magnetic) applications. For instance, Chu and co-workers showed that different shapes of Fe${}_{3}$O${}_{4}$ NPs can destroy cancer cells and tumors in both in-vitro (esophageal cancer cell) and in-vivo (mouse esophageal tumor) models [21].

The mixture of these NPs with a polymer matrix to form nanocomposites can be exploited to develop smart and efficient implants that can potentially overcome the challenges associated with the conventional thermoseeds. These stimuli-responsive materials have played a key role in opening up new frontiers in several fields in biomedical engineering such as drug delivery, tissue engineering, thermotherapy and biosensing [22,23,24]. PDMS, a silicone-based elastomer, is a popular candidate polymer that has been widely used to prepare different nanocomposites for several applications in biomedical engineering due to the possession of an attractive combination of properties such as inertness, bicompatibility, and ease of fabrication [25]. Bonyár and co-workers [26] demonstrated the use of Au/Ag poly-dimethylsiloxane (PDMS) films to enhance the sensitivity of substrates used for surface-enhanced Raman spectroscopy based biosensing applications. Our group used a combination of experiments and models to demonstrate the feasibility of a novel magnetically heated polymer nanocomposite applicator concept for heating of hepatic tumors [27]. Essentially, the applicator was designed as a cannula with a distal heat-generating γ−Fe${}_{2}$O${}_{3}$:PDMS nanocomposite tip and a proximal insulated shaft.

In this study, we explored the feasibility of using photothermally heated superparamagnetic (Fe${}_{3}$O${}_{4}$) nanocomposites (SNCs)-based implants for IT. First, we performed structural and optical characterization of the commercially purchased MNPs. Then, we prepared and studied the structural properties of SNC as a function of weight fraction ($\phi_{\mathrm{mnp}}$ = 10 wt.%). Lastly, cubic implants fabricated from the SNCs were used in photothermal heating experiments in three different media: aqueous solution, cell (in-vitro) suspensions and agarose gel.

## 2. Materials and Methods

### 2.1. SNC Preparation and Implant Fabrication

#### 2.1.1. SNC Preparation

SNC was prepared by mixing Fe${}_{3}$O${}_{4}$ NP with pristine PDMS (Sylgard 184 silicone elastomer kit, Dow Corning Corporation, Auburn, MI, USA). We began with the preparation of the PDMS matrix by mixing the pre-polymer base (monomer) and cross-linking agent (hardener) according to the manufacturer’s recommended ratio of 10:1 by weight [28]. The mixture was stirred for 10 min with a spatula followed by the addition of the Fe${}_{3}$O${}_{4}$ NP and another round of stirring for 10 min to ensure uniform distribution. The resulting uncured mixture was degassed using a desiccator (Vibro-deairator, Gilson Company Inc., Lewis Center, OH, USA) for an hour to remove any trapped air bubbles. Pristine PDMS (Fe${}_{3}$O${}_{4}$-free), was studied as a control. The names of the implant specimens and their compositions are summarized in Table 1.

#### 2.1.2. Implant Specimen Fabrication

The same fabrication process was used to fabricate all samples. It involved the following steps: (a) poured the molten mixture (either SNC or pristine PDMS) into 3D-printed PMMA cubic molds (see Figure 1), which had a cubic geometry with a side length of 0.5 cm, (b) degassed again to eliminate any remaining bubbles before and (c) then placed it into an oven with the temperature set to 100 ${}^{\circ }$C for 35 min according to manufacturer specification [28]. After step (c), we peeled off the implant from the mold and placed them in a desiccator to preserve them.

### 2.2. Materials Characterization Techniques

Fe${}_{3}$O${}_{4}$ NPs (99.7%, 15–20 nm, US Research Nanomaterials Inc., Houston, TX, USA) were characterized by Transmission Electron Microscopy (TEM, Philips CM10, Philips Electron Optics, Eindhoven, The Netherlands), X-ray Diffractomatry (XRD, D8 FOCUS X-ray, Bruker AXS GmbH, Karlsruhe, Germany) at a power of 45 kV × 40 mA and UV-vis spectroscopy (GENESYS 10S UV-vis, Thermo Fisher Scientific, Madison WI, USA) for crystal structure morphology and their absorption at wavelength, λ=810 nm, respectively. The microstructure and the functional groups of the SNCs were studied using Scanning Electron Microscopy (SEM, FEI, Hillsboro OR, USA) and FTIR spectroscopy (Tensor 27, Bruker Inc., Madison, WI, USA) over the wavelength range of 0.6–3 μm${}^{-1}$, respectively. The SEM photographs were obtained at a magnification of 2k and an operating voltage of 10 kV.

### 2.3. Cell Culture and Cytotoxicity Assay

20 μL of 1 × 10${}^{5}$ malignant cell line MDA-MB-231 (American Type Culture Collection, Manassas, VA, USA) was cultured in a 75 cm${}^{2}$ CELLTREAT tissue culture flask (T75 flask), (Pepperell, MA, USA) under normal atmospheric pressure levels in an L-15 medium and supplemented with 100 I.U. mL${}^{-1}$ penicillin/100 μg mL${}^{-1}$ streptomycin and 10% FBS to form a“L15+” medium at 37 ${}^{\circ }$C for 72 h to obtain about 70% confluent cells in the tissue culture flask. Subsequently, the cells were washed with sterile DPBS, followed by Trypsin-EDTA solution to reduce the concentration of divalent cations and proteins that inhibit trypsin action. To detach the cells from the surface of the flask, the solutions in the flask were kept in the incubator for 2 min. L15+ medium was added and the combination of cells in these solutions was centrifuged to allow counting of live cells, and then resuspended in 1 mL of the L15+ medium.

Cytotoxicity was measured with Trypan Blue Exclusion assay according to the manufacturer’s instructions. In brief, the method was used to measure cytotoxicity Trypan blue stock solution was added to cell suspensions and the resulting solutions were loaded into a hemacytometer and examined under an optical microscope at low magnification. Cell viability was determined from the following expression:(1)%cellviability=[1.0−(viablecells÷totalcells)]×100

Viable cells, which are cells that turned blue after trypan blue dye uptake. Total cells represent the number of cells counted before application of the laser. Relative cell viability was measured by comparison with a control, which corresponds to a condition without an implant (cells only).

### 2.4. Agarose Gel Preparation

We prepared the 1.2% agarose gel that used in all experiments by adding 0.6 g of agarose powder was to a beaker containing 50 mL of water. Then we placed the solution in a microwave (Cookworks EM820CFD F-PM 20L Solo, Argos Ltd, Milton Keynes, MK9 2NW, England) and heated at full power (800 W) until agarose was completely dissolved. The hot solution was then removed from the microwave and pour into culture dishes, which were empty or contained MNP-10 implant(s), and allowed to cool and consequently solidify.

### 2.5. Photothermal Measurements

Photothermal heating experiments were performed using three different media: aqueous solution, cell (in-vitro) suspensions and agarose gel (see Figure 2). For each media, irradiation was performed with a near-infrared continuous radiation at 810 nm (Photon Soft Tissue Diode Laser, Zolar Technology & MFG, Mississauga, ON, Canada) with an external adjustable power, $P_{0}$ (0–3 W). The distance between the sample and the laser was 1–2 cm and the laser spot diameter was 1 mm. The $P_{0}$ range that was used was 0.5–1.5 W, step size: 0.5 W. Each sample was identically irradiated for 5 min. To estimate the associated variability, all experiments were run in triplicates.

#### 2.5.1. Aqueous Solution and Cell (In-Vitro) Suspensions

The samples (implants placed in 0.5 mL of deionized water or cell suspension) were contained in a 1.5 mL Eppendorf tube. Deionized water containing no implant was used as control. The resulting temperature rise on the bottom surface of the Eppendorf tube was recorded by thermocouples (K-type, National Instrument, Austin, TX, USA) connected placed to a portable data acquisition system (NI USB-9222A, National Instruments, Austin, TX, USA) and recorded every 30 s with NI-DAQmx (National Instruments, Austin, TX, USA) and software (LabVIEW 8.6, National Instruments, Austin, TX, USA).

#### 2.5.2. Agarose Gel

Three types of samples were prepared including: agarose gel plus (a) zero (studied as control, Figure 3a), (b) one (Figure 3b) and (c) two (interval of 0.5 cm, Figure 3c). We obtained thermal images of the outer top surface of agar using an infrared thermal imaging camera (FLIR 128 System, FLIR Systems Inc., Wilsonville, OR, USA).

## 3. Results

### 3.1. Structural, Optical and Chemical Analysis

#### 3.1.1. MNP Properties

Since the MNPs were purchased commercially, structural characterization was used to verify the specification provided by the manufacturer. The 2θ peaks at 31.5${}^{\circ }$, 35.8${}^{\circ }$, 38.35${}^{\circ }$, 42.75${}^{\circ }$, 47.2${}^{\circ }$, 54.04${}^{\circ }$, 57.24${}^{\circ }$, and 62.75${}^{\circ }$ revealed by X-ray diffraction spectra (Figure 4a) corresponds to diffraction planes 220, 311, 222, 400, 110, 422, 511, and 440, respectively. These planes have been attributed to the cubic spinel phase of Fe${}_{3}$O${}_{4}$ (space group, *Fd-3m*, JCPDS-#19-0629) implying that the NPs are crystalline Fe${}_{3}$O${}_{4}$. TEM revealed the spherical morphology of the MNPs (Figure 4b). Using the Image J software, we analyzed the sizes of about 20 particles (Figure 4b) and found a diameter range, *D*, that was between 13–25 nm (Figure 4c). However, it can be observed that the majority (15 out of 20) of the particles were between the 15-20 nm range as specified by the manufacturer. Under UV-vis-NIR measurement, we observed that 6 mM of the MNPs exhibited an extended optical absorbance that slowly increased in the NIR region compared to the visible light region (see Figure 4d). The UV-vis-NIR absorbance intensity at 810 nm increased linearly with concentration, from 0.35 ([Fe${}_{3}$O${}_{4}$] = 6 mM) to 1.51 ([Fe${}_{3}$O${}_{4}$] = 24 mM) (Figure 4e). The linear increase of absorbance with the range of [Fe${}_{3}$O${}_{4}$] tested in this study is consistent with previously reported results in the literature [29,30]. The absorbance has been attributed to multiple charge (electron) transfer [31]. It is important to note here that at higher [Fe${}_{3}$O${}_{4}$] (100 mM according to Shen et al. [29]) relationship becomes nonlinear because of saturation.

#### 3.1.2. SNC Properties

The functional groups and bonding characteristics of the SNC and pristine PDMS samples were studied and their FTIR spectra are shown in Figure 5a). The bands at I (7.56–8.64 m${}^{-1}$), II (10.10–10.57 m${}^{-1}$), III & IV (12.57 & 14.11 m${}^{-1}$), and V (29.50–29.62 m${}^{-1}$) relate to -CH3 rocking and Si-C stretching in Si-CH${}_{3}$, Si-O-Si stretching, -CH${}_{3}$ deformation in Si-CH${}_{3}$, and asymmetric -CH${}_{3}$ stretching in Si-CH${}_{3}$. It is clear from the results that the characteristic peaks of MNP-0 were not affected when $\phi_{\mathrm{mnp}}$ was 10 (MNP-10). The UV-vis-NIR absorbance intensity of the nanocomposites at 810 nm increased linearly with concentration, from 0.023 ($\theta_{\mathrm{mnp}}$ = 0 wt.%) to 3.7 ($\theta_{\mathrm{mnp}}$ = 10 wt.%) (Figure 5b). The SEM photomicrograph of the surface of MNP-0 shows wrinkles, characteristic mounds and perforations (Figure 5c). Compared MNP-0, MNP-10 showed more wrinkles (Figure 5d) and it can also be observed that the mounds were still present with some trace of MNP clusters, which can affect the optical property and consequently the heat generation capabilities. In a recent study, Wang et al. [33] experimentally found that for the largest aggregate (contained 30 Au NPs) that was used, extinction cross-section of Au NPs reduced by up to 25% resulting in ≈10% decrease in heat generated.

### 3.2. Photothermal Heating

#### 3.2.1. Aqueous Solution

To show Fe${}_{3}$O${}_{4}$ as the main component that provides the majority of the photothermal heat relative to the other components: water and PDMS matrix, we monitored the temperature rise, *T* as a function of time, *t* during the irradiation of (a) water only, (b) water and MNP-0, and (c) water and MNP-10. The results are summarized in Figure 6 ( Error bars are standard error measurements) and Table 2. Generally, it can be clearly seen that the heat generation rates and maximum temperatures obtained after 5 min of deionized water consistently increased with $P_{0}$ and when MNP-0 or MNP-10 implants was added. Starting with the control sample, ΔT increased from ≈2.9 ${}^{\circ }$C to ≈9.0 ${}^{\circ }$C when $P_{0}$ was increased from 0.5 W to 1.5 W (see Table 2). This can be attributed to excitation energy due to the vibrational transitions that are rapidly converted into heat [34]. When the deionized water was irradiated together with implant MNP-0, the temperature increased slightly (relative to water only) by approximately 1.0, 1.67 and 1.8 ${}^{\circ }$C for 0.5, 1.0 and 1.5 W, respectively. The additional heat generated can be attributed to the vibrational overtone of the combination bands of the CH${}_{3}$-groups of the PDMS in the NIR region [35]. When MNP-0 was replaced by MNP-10, maximum ΔT increased by approximately two folds for each three $P_{0}$ relative to that obtained for when MNP-0 was used (see Table 2). For instance, when $P_{0}=1.0$ W was used, ΔT increased to ≈14 ${}^{\circ }$C from ≈7.37 ${}^{\circ }$C (for the MNP-0 implant), representing an increase of ≈90% (see Table 2). The trend of temperature rise recorded here is consistent with results in the literature [21,30]. It is important to know here that we observed higher error margins for the MNP-10 implant. This was expected and can be attributed to non-uniform distributions of the NPs as well as the formation of the NP clusters, which were revealed by the SEM images (see Figure 5d).

#### 3.2.2. Cell Suspensions

Based on the photothermal heating results in aqueous solution, we used a $P_{0}=1$ W and a single MNP-10 implant to generate hyperthermic temperature within MDA-MB cells and investigated the effect of their viability as a function of the duration of irradiation. The results are presented in Figure 7 where cell viability (CV) is presented as a function of $P_{0}$ and *t*. Cell suspensions containing no implant were studied as control.

As expected, the results show that the viability of MDA-MB cells decreased with temperature controlled with $P_{0}$ or *t*. For the case when $P_{0}$ was varied from 0.5 to 1.0 W, the CV for the control sample remained relatively unchanged (≈97%) after 5 min of irradiation. However, when the implant was irradiated together with the cells, the CV dropped to 88.8% and 74.4% for 0.5 W and 1.0 W, respectively (Figure 7a). The corresponding *T* were 40.3 and 47.9 ${}^{\circ }$C (see Figure 7b), respectively. For the case when *t* was varied, the CV decreased from 74.4% to 56.4% when the time was increased from 5 to 10 min under irradiation power of 1.0 W (Figure 7c). The corresponding corresponding final temperatures were 47.9 and 51.7 ${}^{\circ }$C (Figure 7d), respectively. The time dependence of the cell death due to elevated temperature levels has been previously reported [36,37].

#### 3.2.3. Agarose Gel

A desirable feature of the implant technique is the opportunity to control the geometry of tissue lesion by varying the number as well as the arrangement of implants within the tissue [4,5]. We studied this feature using an 1.2% agarose gel model to represent soft tissue [38].

When agarose gel was irradiated without an implant, ΔT was relatively small ($\Delta T3{\;}^{\circ }$C) resulting in a non-distinctive temperature profile (Figure 8a). For the case when a single implant was heated together with the agarose gel, ΔT increased by about 333% ($\Delta T=13{\;}^{\circ }$C) and the temperature profile was revealed to have circular geometry and the maximum temperature ($\Delta =42{\;}^{\circ }$C) occurred around the implant and decreased radially outward (Figure 8b). In addition to the obvious increase in ΔT (about ≈443% and ≈25%) when the number of implants was doubled, the temperature distribution increased and took the form of an ellipsoidal shape (Figure 8c). The results demonstrate how the temperature profile can potentially be manipulated using the number and configuration of the implants within the tumor. These results are similar to previously reported studies using thermoseeds [4,5].

## 4. Discussion

The preclusion of the clinical translation of the conventional thermoseed technique have been attributed to two main problems: (a) biocompatibility and corrosiveness of the metallic alloys and (b) the high number of implants needed to achieve therapeutic temperature levels due to the inability of the AMF to directly heat the tissue. The combination of multifunctional polymer nanocomposites and photothermal heating in the manner present here has the potential to overcome the problems associated with the thermoseed technique.

For the problem related to the material properties of metallic alloys, implants fabricated from multifunctional nanocomposite will benefit from the plethora of novel biocompatible polymer systems and nanomaterials coupled with their facile synthesis methods [25]. In the last few decades, the design, fabrication and application of these materials have played a major role in opening up new frontiers in several theranostic applications [23,24,25]. Some of the most commonly used polymeric implants include silicone rubbers, polyethylenes, polyetheretherketones (PEEK) and bioabsorbables-polylactic acid (PLA), polyglycolic acid (PGA) and their copolymers and Teflon [25,39,40]. Unlike the conventional metallic alloys thermoseeds, these polymers are mostly biocompatible, resistant to corrosion as well as amenable to additive fabrication techniques such as 3D printing that can enable rapid prototyping and fabrication of implants at the point of care. The desirable thermal self-regulating feature of ferromagnetism can be introduced into the polymer using purposely designed metallic oxide NPs, which have been previously reported in the literature [41,42]. Furthermore, the efficacy and safety of treatment can be enhanced by exploiting properties of polymer and NPs. One option is using specialized polymer systems to design implants for multimodal (simultaneous heat + drugs) applications [23,39]. Another option is the exploitation of the capability of metal-oxide NPs to generate heat under NIR laser and AMF exposure to design implants for the so-called DUAL-mode applications, which have been shown to enhance the safety of treatment [30,43].

When NIR radiation interacts with biological media, it can generate heat, unlike AMF, which has no thermal effect. This phenomenon can be exploited to solve the thermoseed technique’s problem of requiring a high number of implants required to achieve therapeutic temperature levels. The optothermal response of any material depends on an interplay between the optical properties of the material (absorption/scattering coefficient, anisotropy) and the radiation protocol, which includes optical parameters of the incident light (wavelength, power/energy, spot size), irradiation time and mode of laser delivery (extracorporeal or interstitial applicator) [44]. Although NIR radiation has been shown to have good penetrability in biological media [45], issues related to the turbidity of biological media affect the light distribution and limit the applicability of extracorporeal irradiation to superficial tumors [46]. Recent technological advances in laser delivery techniques have led to the development of several applicators that can be exploited to enhance the efficiency of light delivery to deep-seated tumors [14]. These minimally invasive applicators, which are usually light-guiding optical fiber waveguides with diffusing tips [14,47], have been described in the literature as safe and effective [48].

Finally, the results that were presented here offer a context to discuss the feasibility of using photothermally-heated nanocomposite implants, However, an extensive study that combines experiments and computational techniques is needed to obtain a realistic assessment of the actual performance of this novel approach [49,50].

## 5. Conclusions

In this paper, we explored the properties of superparamagnetic polymer nanocomposites in terms of their structural, optical and photothermal capabilities. After confirming the properties of Fe${}_{3}$O${}_{4}$ NPs, we mixed them with molten PDMS to form the nanocomposite, poured the mixture into cubic molds and cured at 100 ${}^{\circ }$C for 35 min. After curing, the resulting nanocomposite implant had no new bond formed between the NP fillers and PDMS matrix. Subsequently, we elucidad the photothermal heating behavior of the implants in 3 different media: aqueous solution, cell suspensions (in-vitro) and agarose gel. Our measurements in aqueous solution clearly showed that the addition of the Fe${}_{3}$O${}_{4}$ NPs to the PDMS matrix increased the temperatures. Similar trends were observed for cell suspensions and agarose gel to reduce cell viability and lesion size, respectively. Collectively, the results demonstrate that SNC implants embedded in a media can generate photothermal heat to increase temperature levels when irradiated with NIR laser. Furthermore, SNCs provide several opportunities for the potential replication of desirable features associated with conventional thermoseed as well as overcome challenges associated with them towards the development of a smart and efficient implant-based technique for IT. Our long-term goal is to develop an IT technique based on polymer nanocomposite implants that can be optimized based on patient data and fabricated at the point of care.

## Figures and Tables

**Figure 1 nanomaterials-12-00955-f001:**
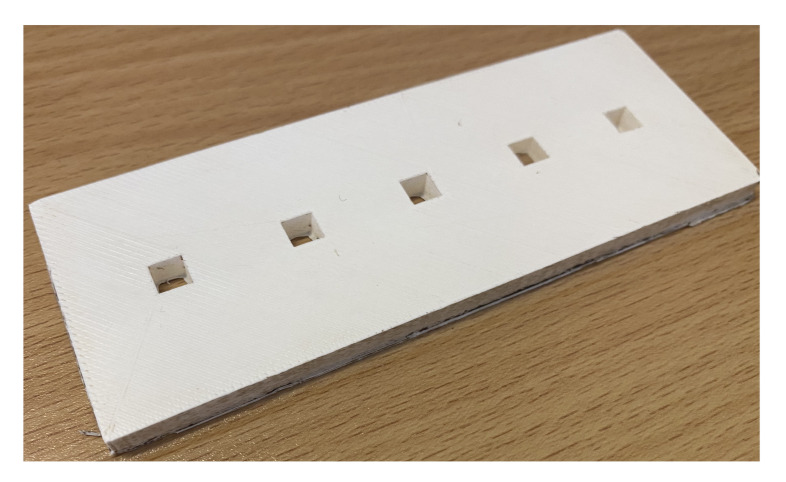
Photograph of the 3D-printed PMMA mold used to fabricate the implant specimens.

**Figure 2 nanomaterials-12-00955-f002:**
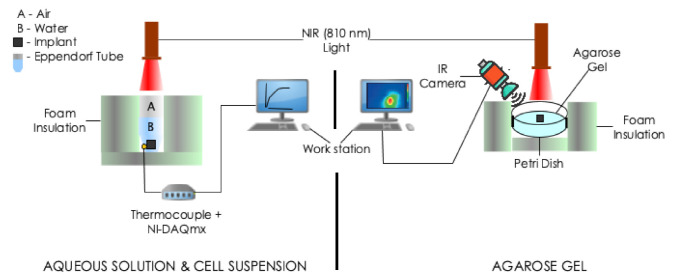
Schematic of the experimental setups for photothermal measurments. The experiments involved the irradiation of implant(s) embedded in: (**Left**) 0.5 mL deionized water or cell suspension contained in a 1.5 mL Eppendorf tube and (**Right**) agarose gel contained in a plastic Petri dish with a different power of NIR radiation at λ=810 nm. Temperature measurements were taken with a thermocouple or IR camera.

**Figure 3 nanomaterials-12-00955-f003:**
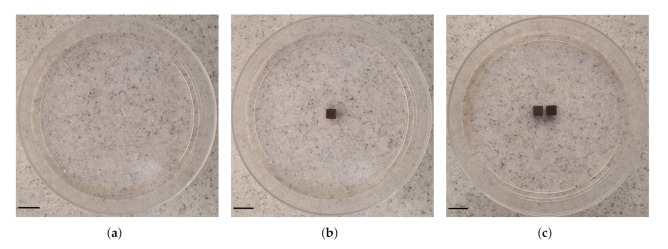
Photographs of the configuration of MNP-10 implant(s) in agarose gel. Three samples were studied: agarose gel and (**a**) no (studied as control), (**b**) one and (**c**) two (interval between them is 0.5 cm) implant(s). The samples were placed into 90 mm cell culture dishes. Scale bar: 1.0 cm.

**Figure 4 nanomaterials-12-00955-f004:**
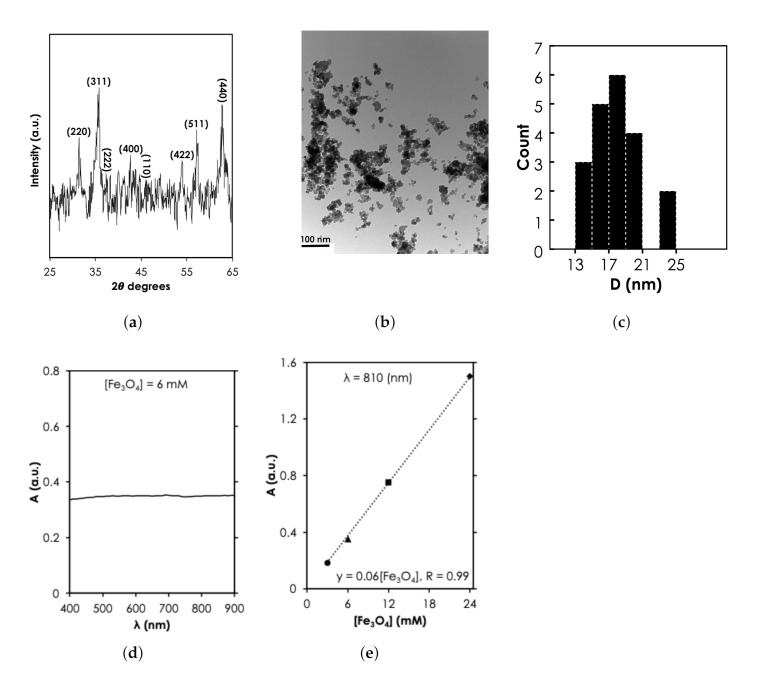
Structural and optical characterization of MNPs results. (**a**) X-ray diffraction spectra of MNPs at a power of 45 kV × 40 mA (Adapted from [32]). (**b**) Transmission electron microscopy of Fe${}_{3}$O${}_{4}$ at magnification of 0.5 mm, scale bar: 100 nm and (**c**) the corresponding nanoparticle count as a function of size, D. UV-vis-NIR absorbance spectra as a function of (**d**) λ for 6 mM of MNPs and (**e**) [Fe${}_{3}$O${}_{4}$] at λ = 810 nm.

**Figure 5 nanomaterials-12-00955-f005:**
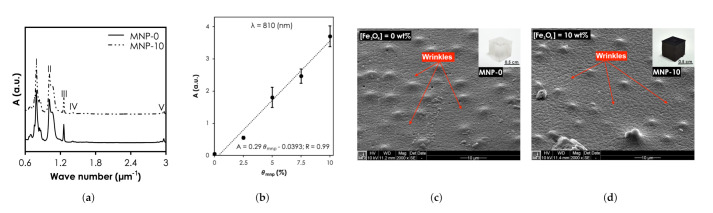
Structural and optical characterization SNCs results. (**a**) Fourier Transform Infrared spectra of all samples. (**b**) The UV-vis-NIR absorbance intensity of the nanocomposites at 810 nm. Scanning Electron Microscopy of (**c**) MNP-0 and (**d**) MNP-10.

**Figure 6 nanomaterials-12-00955-f006:**
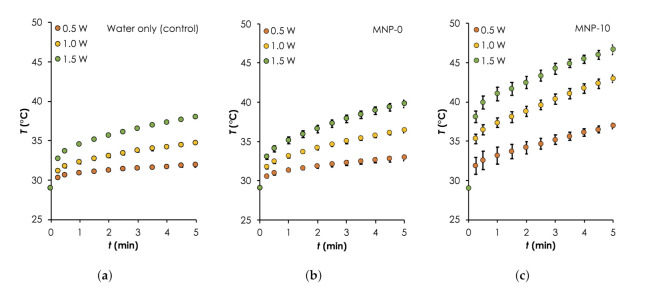
Photothermal heating generation in water results. A comparison of temporal curves for the different samples: (**a**) Water only (control), as well as that consisted of implants (**b**) MNP-0 and (**c**) MNP-10 and deionized water during irradiation with different powers: $P_{0}$ (0.5–1.0 W, step size: 0.5 W, λ=810 nm) for 5 min. For all measurements, $T_{0}=29$${}^{\circ }$C.

**Figure 7 nanomaterials-12-00955-f007:**
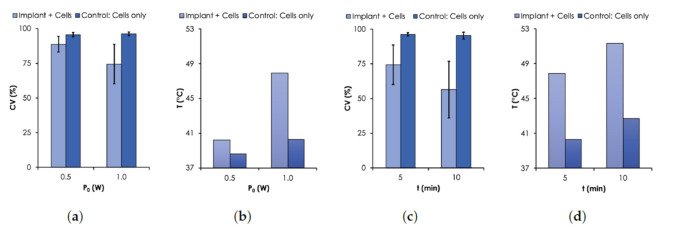
Cell Viability (CV) results. CV was assessed with Trypan Blue assay using MDA-MB-231 breast cancer cells. (**a**) CV and (**b**) *T* as a function of $P_{0}$ (0.5–1.0 W, λ=810 nm). (**c**) CV and (**d**) *T* as a function of time. For all measurements, an initial $T_{0}=37$${}^{\circ }$C.

**Figure 8 nanomaterials-12-00955-f008:**
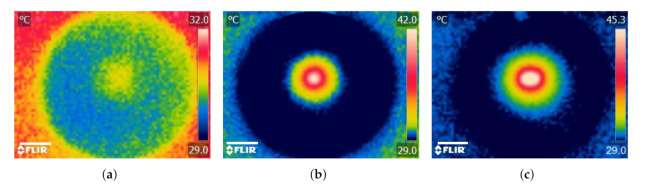
Photothermal heating generation in agarose gel results. Infrared thermographic images after NIR irradiation ($P_{0}=1.0$ W, λ=810 nm) of (**a**) agarose gel only (control) and agarose gel containing (**b**) a single and (**c**) double MNP-10 implants. Scale bar: 1.5 cm.

**Table 1 nanomaterials-12-00955-t001:** Composition of the implant specimens.

No.	Sample Name	Weight Fraction, (wt.%)	
		[Fe${}_{3}$O${}_{4}$], (Magnetite)	[n−C${}_{2}$H${}_{6}$OSi−n], (PDMS)
		$\phi_{\mathrm{mnp}}$	$\phi_{\mathrm{pdms}}$
1	MNP-0	0	100
2	MNP-10	10	90

**Table 2 nanomaterials-12-00955-t002:** Summary of the mean ΔT (±s.e.m) (${}^{\circ }$C) measured after 5 min of irradiation of the 4 different samples, which included water only (a) as well as those consisting of implants: MNP-0 (b), MNP-5 (c) MNP-10 (d) and deionized water during irradiation with different laser powers: $P_{0}$ (0.5–1.0 W, step size: 0.5 W) for 5 min.

$P_{0}$ (W)	ΔT(±s.e.m.) (${}^{\circ }$C)		
	Water Only (Control)	MNP-0	MNP-10
0.5	2.9 (±0.3)	3.9 (±0.6)	7.9 (±0.6)
1.0	5.7 (±0.2)	7.37 (±0.4)	14.0 (±2.2)
1.5	9.0 (±0.1)	10.8 (±0.8)	17.7 (±1.6)

## Data Availability

The data presented in this study are available in this article.

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
