# Peer review of "Photothermally-Heated Superparamagnetic Polymeric Nanocomposite Implants for Interstitial Thermotherapy"

_nanomaterials, 2022, doi:10.3390/nano12060955_

Round 1

Reviewer 1 Report

This work developed explored a superparamagnetic polymer nanocomposite which showed good optical and photothermal capabilities. The SNC implants generated photothermal heat that increased the temperature of deionized water to different level at different rates, decreased the viability of MDA-MB-231 cells and regulated lesion size in agarose gel as a function of laser power only, laser power or exposure time and the number of implants, respectively. This work is interesting and the results are useful. It is recommended for major revision and the following questions should be addressed.

In the materials characterization, it is claimed that “Fe 3 O 4 NPs (99.7 %, 15-20 nm, US Research Nanomaterials Inc., Houston, TX, USA) were 107 characterized by Transmission Electron Microscopy (TEM, Philips CM10, Philips Electron 108 Optics, Eindhoven, The Netherlands) at a magnification of 0.5 mm.” can the author explain the clarification “at a magnification of 0.5 mm”. What is the meaning of this description?

Did the author treat the surface of the bough Fe3O4? Can they be well dispersed within the PDMS?

Many misused symbols are found in the manuscript, they should be carefully checked.

Why the absorption of Fe3O4 cannot be observed in the FTIR? The weight fraction of Fe3O4 is 10%, which is high enough to be detected by FTIR.

The reference should be formatted carefully.

Reviewer 2 Report

The authors have assessed SNC-PDMS and MNP implant for IT. The authors have studied the structural properties of SNC and their photothermal heating capabilities of the implants in three different media. The authors have discussed the opportunities for the development of a smart and efficient strategy to enhance the efficacy of conventional interstitial thermotherapy. Overall, this work can inspire more material design ideas for photothermally-heated polymer-based SNC implant technique. Therefore, I would like to recommend this work to publish in Nanomaterials. Below are a few suggestions for the authors.

  1. For Figure 4b, TEM image of Fe3O4 is cover by the inset. It is not easy to see. The inset should be separately provided.
  2. For Figure 4c, the raw spectra of MNP-0, MNP-10 should be provided to verify their absorptions.
  3. The authors have mentioned “Compared MNP-0, MNP-10 showed more wrinkles”. The authors should provide scientific data or indicate in the figure.
  4. For Figure 7, the standard deviation (SD) should be provided for the result of cell viability.
  5. Iron-oxide NPs have been approved by the FDA for clinical magnetic thermotherapy but not for photothermal therapy. Therefore, the reason for the use of iron-oxide NPs in photothermally-heated superparamagnetic polymeric nanocomposite implants could be emphasized in the introduction.
  6. For the introduction “Minimally invasive photothermal therapy has recently received a lot of attention because......”, more references could be cited to broaden the introduction. (https://doi.org/10.3390/nano10061123)

Round 2

Reviewer 1 Report

The article can be accepted after well preparing the image. (Figure 8)

Author Response

Our thanks go to the reviewers for their comments. The paper has been modified to address the comments of the reviewers.  Responses are in red text.

  1. The article can be accepted after well preparing the image. (Figure 8)

Our apologies. Fig. 8 has been placed in the right position.